# Regional disparities in subjective wellbeing across Europe: A fuzzy hybrid TOPSIS approach

**Alessandro Indelicato**[1]*, **Juan Carlos Martín**[1], **Vincenzo Marinello**[2]

**1** Department of Analysis of Applied Economics, Universidad de Las Palmas de Gran Canaria, Las Palmas de Gran Canaria, Spain, **2** Department of Economic and legal sciences, University of Enna "Kore", Enna, Italy

* alessandro.indelicato@ulpgc.es

## Abstract

The study analyses differences in subjective well-being (SWB) across European regions. Through the Fuzzy-Hybrid TOPSIS approach, we analyse SWB at the country and regional levels. Additionally, a quantile regression model is employed to investigate the impact of socio-economic factors on SWB. The International Social Survey Programme (ISSP) dataset from 2017 is used for seven countries: Denmark, Germany, Spain, France, Finland, Hungary and Slovenia. The synthetic indicator is derived from four indicators: happiness, life satisfaction, goal achievement, and family pressure. At the country level, Germany achieves the highest SWB score (0.69), while Hungary records the lowest (0.51). Regional analysis shows German regions (particularly Saarland and Schleswig-Holstein) and Spanish regions (notably La Rioja and Baleares) occupy top positions in the SWB rankings. Quantile regression results confirm that age, education level, and income significantly influence SWB, with older individuals, those with intermediate education levels, and higher-income earners showing consistently higher SWB values.

## 1. Introduction

Based on previous studies [1,2,3,4,5], we measure the SWB across different European regions. To create this indicator, the study uses the 2017 ISSP social network module. Recognising that survey data are uncertain, the study employs a Fuzzy-Hybrid TOPSIS approach to develop the SWB indicator. Then it applies quantile regression to analyse differences among individuals with low, medium, or high SWB with respect to factors such as age, education, income, and religiosity. The study examines seven European Union (EU) countries: two in Northern Europe (Denmark and Finland), two in Central Europe (France and Germany), two in Eastern Europe (Hungary and Slovenia), and one in Southern Europe (Spain). Other EU countries cannot be included due to data limitations in the ISSP module.

**Data availability statement:** All data underlying the findings are openly available in the ISSP Data Archive at GESIS – Leibniz Institute for the Social Sciences, with DOI doi.org/10.4232/1.13322.

**Funding:** Dr Alessandro Indelicato research is funded by the research fellowship "Catalina Ruiz," provided by the Consejo de Economía, Conocimiento y Empleo of the Gobierno de Canarias, the Agencia Canaria De Investigación Innovación Y Sociedad De La Información (ACIISI), and Fondo Social Europeo of the EU, through the Universidad de Las Palmas de Gran Canaria (Spain).

**Competing interests:** The authors have declared that no competing interests exist.

The year for which the data was collected is still relevant for this work, as the World Happiness Index shows that well-being rankings in Europe have not changed much over time. Comparing the 2017 and 2024 reports, the rankings of Denmark, Finland, Germany, France, Spain, Slovenia, and Hungary have changed only slightly, and no significant shifts have occurred that would invalidate cross-country or cross-area comparisons. These links are well established in many SWB studies, which consistently demonstrate that trends in well-being are gradual and that variations in SWB scores across countries have remained stable over time. Therefore, using data from 2017 does not invalidate the results, particularly since this study focuses on current conditions and regional differences rather than year-on-year changes.

The remainder of the paper is structured as follows: Section 2 provides a brief literature review; Section 3 illustrates the data; and Section 4 details the adopted methodology. Section 5 presents the results, and Section 6 discusses them. The paper concludes with Section 7, which provides final remarks.

## 2. Brief theoretical background

Since the early 1900s, researchers have attempted to quantify well-being and the societal factors that influence it [6]. Subjective well-being (SWB) has been a particular focus of research [7,8,9, Jean-Paul & Martine, 2018]. The OECD has developed methods to measure SWB, emphasising the importance of understanding how people evaluate their lives, emotions and growth [10,11].

Studies suggested that SWB is related to happiness [12] and life satisfaction [13]. In this context, Diener et al. [14] argued that happiness is an important factor in SWB and that people's feelings can change. However, other studies suggest that well-being comprises many aspects, meaning that if we use only one measure, such as life satisfaction, we may miss differences between countries or regions. For instance, regions that appear to have similar SWB may differ in terms of health, purpose, relationships, or community [15].

Over the past ten years, studies have examined variations in SWB across Europe. Rather than simply considering country-level data, researchers are examining how different regions and localities influence SWB [16,17] . One such difference is between cities and rural areas. Previous studies across the EU have suggested that people in rural areas are happier [18] . However, more recent data from the UK indicates that city inhabitants may experience poorer well-being and be less satisfied with society [19]. Thus, these results demonstrate how perception of life satisfaction is influenced by geographical location.

Moreover, it is worth noting that there is a U-shaped relationship between age and SWB, with people feeling at their worst in middle age [20]. However, new evidence challenges this concept. Blanchflower et al. (2025) [21] claim that the U-shaped curve has largely disappeared in Western Europe, and that in certain southern European countries, people become less satisfied with life as they age. Education still affects SWB [22], as does income, although its significance diminishes after a certain point [23][.

The people around you matter as well as what you do. Feeling part of a group and trusting others have always been linked to happiness [24,25]. Perceiving social justice also contributes to happiness; people are happier in countries where they feel that things are fair [26]. These results suggest that the feelings are tied to society.

Thus, a growing body of research suggests that subjective well-being (SWB) is influenced not only by financial factors but also by a region's overall performance. For example, one study of 228 European regions found that higher SWB was associated with higher economic output, with even a slight increase in SWB leading to an increase in economic output [27]. Regions where people are happier are more likely to attract workers, helping them generate ideas and grow. These findings are consistent with Fredrickson's broaden-and-build theory [28], which proposes that positive emotions broaden individuals' perspectives, facilitating the development of new skills and capabilities over time.

## 3. Data

The Social Network module of the International Social Survey Programme (ISSP) (2017) has been used in the study. This module provides four items for measuring SWB [3]. In these four items, the respondents in the ISSP questionnaire provide measures of happiness, life satisfaction, the ease of accomplishing goals, and family pressure. The relative questions of these items are respectively worded as follows:

During the past four weeks, how often have you felt unhappy and depressed?

All things considered, how satisfied are you with your life as a whole nowadays?

It is easy for me to accomplish my goals.

In general, do your family members put pressure on you about the way you live or organise your personal life?

The responses to items 1 and 4 are based on a 5-point semantic scale, where one stands for no, never, two for yes, but rarely, three for yes, sometimes, four for yes, often, and 5 for yes, very often. Answers to items 2 and 3 are based on a 7-point semantic scale according to (1) completely satisfied or completely true, respectively; (2) very satisfied or mostly true; (3) fairly satisfied or somewhat true; (4) neither satisfied nor dissatisfied or neither true nor untrue; (5) fairly dissatisfied somewhat untrue; (6) very dissatisfied or mostly untrue; and (7) completely dissatisfied or completely untrue. In order to obtain a synthetic indicator that measures well-being, the answer values were all reversed, resulting in higher values aligned with higher well-being.

The study considers 7 European Union countries. The country selection was mainly based on having a territorial representation of different areas: two countries represent the Northern EU, Denmark and Finland; two countries represent the Central-Western EU, France and Germany; two countries have been chosen to represent the EU Eastern area, Hungary and Slovenia; and finally, one country in the representation of the EU Southern area, Spain. Unfortunately, we could only select one country to represent the South EU zone because the items were not included in any other Southern European country. The countries are initially selected to provide an overall representation of the EU's different territories, thereby making the empirical problem more tractable. The seven selected countries enable researchers to analyse SWB across 179 regions, which are considered an effective proxy for extracting interesting insights from other socio-economic drivers to explain SWB. Balancing sample representativeness and empirical tractability is always challenging. In the current study, however, we opted for caution, using only a limited number of EU countries to ensure that the results could be more effectively addressed and explained.

The dataset contains 9130 respondents. Table 1 shows that more respondents are from Germany (18.63%) and Spain (18.98%), and that the sample is more represented by women (52.53%). The respondent profile regarding age is 46–55 (19.09%) or 56–65 (19.29%). The majority of the sample have lower secondary (22.74%), upper secondary (17.45%) or post-secondary (18.00%) education levels. Regarding religion, 36.96% of the sample are Catholics, 25.51% are Protestant, and almost 31% are agnostic. Most respondents (41.58%) never attend religious events, while only 1.26% attend them several times a week. The sample's income perception is medium-high, but only 0.62% report the highest income. Finally, the general health status of the respondents is good (38.48%) or very good (28.65%).

The ISPP dataset provides information at the country and regional levels. This study analyses SWB at the country and NUTS2 levels. France provides data disaggregated at the NUTS3 level, i.e., in this country, the SWB indicator is given at the provincial level. NUTS is the acronym for the nomenclature of territorial units for statistics used in the EU, where NUTS2

**Table 1. Sample composition by country and socio-demographic characteristics.**

| Group | n | % | Group | n | % |
|---|---|---|---|---|---|
| Denmark | 1079 | 11.82% | No religion | 2829 | 30.99% |
| Finland | 1074 | 11.76% | Catholic | 3374 | 36.96% |
| France | 1489 | 16.31% | Protestant | 2329 | 25.51% |
| Germany | 1701 | 18.63% | Orthodox | 112 | 1.23% |
| Hungary | 1007 | 11.03% | Other-Christian | 49 | 0.54% |
| Slovenia | 1047 | 11.47% | Jewish | 14 | 0.15% |
| Spain | 1733 | 18.98% | Buddhist | 146 | 1.60% |
| Male | 4361 | 47.77% | Hindu | 16 | 0.18% |
| Female | 4769 | 52.23% | Other Asian Religions | 4 | 0.04% |
| <=24 | 662 | 7.25% | Other religions | 38 | 0.42% |
| 25-35 | 1280 | 14.02% | Several times a week or more often | 115 | 1.26% |
| 36-45 | 1493 | 16.35% | Once a week | 513 | 5.62% |
| 46-55 | 1743 | 19.09% | 2 or 3 times a month | 329 | 3.60% |
| 56-65 | 1761 | 19.29% | Once a month | 259 | 2.84% |
| 66-75 | 1383 | 15.15% | Several times a year | 1374 | 15.05% |
| >75 | 716 | 7.84% | Once a year | 836 | 9.16% |
| No formal education | 207 | 2.27% | less frequently than once a year | 1717 | 18.81% |
| Primary school | 586 | 6.42% | never | 3796 | 41.58% |
| Lower secondary | 2076 | 22.74% | Lowest Income | 156 | 1.71% |
| Upper secondary | 1593 | 17.45% | Income 2 | 222 | 2.43% |
| Post-secondary | 1643 | 18.00% | Income 3 | 590 | 6.46% |
| Lower-level tertiary | 1617 | 17.71% | Income 4 | 881 | 9.65% |
| Upper lever tertiary | 1325 | 14.51% | Income 5 | 2280 | 24.97% |
| Poor (health) | 389 | 4.26% | Income 6 | 1906 | 20.88% |
| Fair (health) | 1579 | 17.29% | Income 7 | 1582 | 17.33% |
| Good (health) | 3513 | 38.48% | Income 8 | 895 | 9.80% |
| Very good (health) | 2616 | 28.65% | Income 9 | 158 | 1.73% |
| Excellent (health) | 935 | 10.24% | Highest Income | 57 | 0.62% |

N: number of individuals; %: percentage.

are the primary regions for applying regional policies, and NUTS3 are smaller regions, or provinces, for specific diagnoses. We used NUTS3 data for France because that was available from the ISSP; for other countries, we used NUTS2 data. Admittedly, this means the territorial scales differ slightly, but the comparison is still valid. Our focus is on how regional differences stack up and on the ranking of well-being across countries. Furthermore, since NUTS3 regions are contained within NUTS2 regions, having different levels of detail does not affect the model's ability to determine how SWB changes across countries.

## 4. Methodology

### 4.1 Fuzzy-hybrid TOPSIS

**4.1.1 Fuzzy set theory.** The information provided by questionnaires, such as that of ISSP, is vague and subjective. The responses are partly constrained to individual and inaccurate judgments [29] and, therefore, cannot be synthesised through simple statistical tools, such as the average [30]. The Fuzzy Set Theory (FST) is an approach that has been applied across several research fields, including green energy [31], logistics [32], and social science [33,34].

FST is a practical approach because it can manage the vagueness of the information [35]. Vagueness is a phenomenon that runs through people's thinking and language. This information is flexible and can be manipulated by making subtle changes to the wording of the questionnaire topics. There is a relatively fixed and well-defined set of linguistics for any real value. On the other hand, terms expressed in vague terms lack well-defined rules. Thus, a fuzzy approach can convert raw information into crisp values [36].

Zadeh [37] set up the basis of a new methodology able to study complicated systems. He has generalised the idea of the classic logic "true" or "false" to an interval [0, 100] that represents a world of discourse in a fuzzy environment, using real numbers. Thus, while the classic logic can only report two results (1, true, or 0, false), the fuzzy logic expresses an associated membership function for each element, which indicates to what extent the element is part of that fuzzy set.

Let $A$ be a fuzzy set in $X$, $\mu_A(x)$ is the membership function $A : X \rightarrow [0, 1]$ known in a universe of discourse $X \in [0, 100]$. The closer is x is to 1, the higher is the intensity of $x$ to belong to $A$ [38,35]. Given $\mu_A(x)$ the membership function used to measure the relative truth in the statements $x \in A$ [39,37]. $X$ express the fuzzy set of the universe of discourse that is a generalisation of the classical (true false) logic.

Thus, the vagueness of information provided by the questionnaire is managed by FST. First, the information is first transformed into Triangular Fuzzy Numbers as a useful tool to manage the vagueness [40]. Thus, for each item, the responses of respondents are transformed into a 3-uple $(a_1, a_2, a_3)$, The triplet provide intrinsically the membership function of the TFN as follows:

$$\mu_A(x) = \begin{cases} \frac{x-a_1}{a_2-a_1} & a_1 \leq x \leq a_2 \\ \frac{x-a_3}{a_2-a_3} & a_2 \leq x \leq a_3 \\ 0 & otherwise \end{cases}$$

In order to not lose generality, each point of both 5-point and 7-point semantic scales has been assigned a TFN [39]. The triplet for each point of both scales has been chosen according to previous literature [41,42]. Table 2 shows the TFNs used in the study for both semantic scales. Thus, fuzzy logic adequately manages the information's vagueness because all consecutive TFNs overlap [43].

This approach can be used at the individual and aggregate levels. Thus, the analysis can be performed for each group of research interest, and, again, the Fuzzy Set Logic Algebra provides an excellent environment to obtain the aggregate TFNs average. These aggregate values are given by:

$$\widetilde{A} = (a_1, a_2, a_3) = \left(\frac{1}{n}\right) \otimes \left(\widetilde{A}_1 \oplus \widetilde{A}_2 \oplus ... \oplus \widetilde{A}_n\right) = \left(\frac{\sum_{i=1}^{n} a_1^{(i)}}{n}, \frac{\sum_{i=1}^{n} a_2^{(i)}}{n}, \frac{\sum_{i=1}^{n} a_3^{(i)}}{n}\right)$$

**Table 2. Triangular fuzzy numbers.**

| Scale | TFN[a] | TFN[b] |
|---|---|---|
| 1 | (0,0,30) | (0, 0, 10) |
| 2 | (20,30,40) | (0, 10, 30) |
| 3 | (30,50,70) | (10, 30, 50) |
| 4 | (60,70,80) | (30, 50, 70) |
| 5 | (70,100,100) | (50, 70, 90) |
| 6 | | (70, 90, 100) |
| 7 | | (90, 100, 100) |

TFN: Triangular Fuzzy Number; [a]: Indicators 1 and 4; [b]: Indicators 2 and 3.

Where $\otimes$ is the multiplication of a scalar and a TFN, and $\oplus$ stands for the internal addition of TFNs [44]. Thus, obtaining a TFNs matrix in which each element summarises the information at the group level is possible. This matrix is complicated to analyse because the elements contain much information. For this reason, TFNs are usually defuzzified into real numbers that contain crisp and clarified values. This defuzzification is given by:

$$V_{\widetilde{A}} = \frac{(a_1 + 2a_2 + a_3)}{4}$$

The Fuzzy approach is well-suited to the nature of SWB data. SWB data, which is gathered using ordinal semantic scales, often includes uncertainty, unclear language and personal opinions. Rather than treating ordered categories as equally spaced numbers, Fuzzy Set Theory recognises that people's thoughts and understanding of scale points are not straightforward and can overlap. As illustrated in previous applications, consecutive TFNs overlap systematically, reflecting the empirical vagueness inherent to survey-based constructs [33]. This property is critical when analysing social constructs for which there is no unique objective function to measure latent concepts, making fuzzy logic superior to purely numerical aggregation techniques.

**4.1.2 TOPSIS.** Fuzzy algebra addresses uncertainty, and the technique for order of preference by similarity to the ideal solution (TOPSIS) provides a multicriteria ranking that aligns with the objective of creating a synthetic indicator. Unlike the Structural Equation Model (SEM), the fuzzy hybrid TOPSIS (FHT) does not require the strict assumptions of latent-variable methods, such as full or partial scalar invariance, which are rarely applicable in cross-group studies and can result in skewed or incomparable outcomes, as demonstrated by Indelicato & Martín [33]. FHT is a more adaptable and empirically sound choice for SWB comparisons across countries and regions than SEM because it does not require equal constraints across groups.

TOPSIS is one of the most commonly used multicriteria methods for fuzzy sets when dealing with vagueness and subjective information [40]. TOPSIS applied to fuzzy sets is a hybrid approach widely used in academia to rank a set of possible alternatives [Li, 2007, 45].

The FHT synthetic indicator is obtained following three different stages [46]. First, the Positive Ideal Solution (PIS) and Negative Ideal Solution (NIS) are calculated. PIS is expressed by the maximum defuzzified value for each group and each item, while NIS is the minimum value in the defuzzified matrix for each item and group [47]. Both positive PIS and NIS are calculated as follows:

$$A_j^+ = \left\{ (maxV_{ij}), j = 1, 2, \ldots, J \right\}, i = 1, 2, \ldots m$$

$$A_j^- = \left\{ (minV_{ij}), j = 1, 2, \ldots, J \right\}, i = 1, 2, \ldots m$$

where $i = 1$ to m (groups), $j = 1$ to J (criteria), and $V_{ij}$ are the crisp values of the defuzzified matrix. As in [48], all criteria are considered a benefit because higher indicator values represent higher subjective well-being. Afterwards, the Euclidian distances between each group and PIS and NIS are calculated as follows:

$$S_i^+ = \sqrt{\sum_{j=1}^{J} \left( A_j^+ - V_{ij} \right)^2}$$

$$S_i^- = \sqrt{\sum_{j=1}^{J} \left( A_j^- - V_{ij} \right)^2}$$

Thus, the synthetic indicator that measures the subjective well-being of the groups is given by:

$$SWB_i = \frac{S_i^-}{S_i^+ + S_i^-} \rightarrow [0, 1]$$

Higher scores on the indicator mean higher SWB. Since all the different components determine the synthetic score, it is useful to examine how much each component contributes to the overall score. As in Martín and Indelicato [34], we calculate elasticities to determine how each factor affects the SWB score within each group. Although elasticities are not usually part of TOPSIS, they are increasingly being used in fuzzy-hybrid MCDM studies to show how changes in one factor affect the overall score. For each group i and attribute j, the elasticity is calculated as follows:

$$\eta_{ij} = \frac{\Delta\%SWB_i}{\Delta\%V_{ij}}$$

where $SWB_i$ denotes the synthetic TOPSIS score of group $i$ and $V_{ij}$ represents the corresponding defuzzified value of item $j$. This shows the percentage change in the SWB score resulting from a one percent change in one of the factors. It provides a straightforward method of determining the impact of each aspect of well-being on the overall score.

### 4.2 Quantile regression

This study aims to analyse how socio-economic characteristics can shape SWB. In this context, quantile regression can provide an overview of which covariate influences more or less the individual SWB. Quantile regression models is an effective tool to study the influence of a set of variables on other, using quantiles of the conditional distribution of the dependent variable as functions of covariates [49].

Let $y = \boldsymbol{x}'\boldsymbol{\beta} + \varepsilon$ be a linear regression model. The quantile regression model is given by the following assumption:

$$Q[y|\mathbf{x}, q] = \mathbf{x}'\beta q + \varepsilon$$

$$suchthat$$

$$Prob[y \leq \mathbf{x}'\beta q|\mathbf{x}] = q$$

$$0 < q < 1$$

where $q$ is the $q_{th}$ quantile. If $th$ is equal to 0.5, it is the median quantile. $\varepsilon$ is the error term.

The model provides richer results because the specified coefficients are indexed by $q$. Since $q$ can vary between 0 and 1, there are an infinite number of possible parameter vectors [50].

Let $b_q$ the estimator of $\beta_q$ in the $q_{th}$ quantile. The estimator is calculated by minimising the following function:

$$F_n(\beta_q|y, \mathbf{x}) = \sum_{i=y\leq\mathbf{x}_i'\beta_q}^{n} q|y_i - \mathbf{x}_i'\beta_q| + \sum_{i==y\leq\mathbf{x}_i'\beta_q}^{n} (1-q)|y_i - \mathbf{x}_i'\beta_q| = \sum_{i=1}^{n} g(y_i - \mathbf{x}_i'\beta_q|q)$$

where $g(e_{i,q}|q) = \begin{cases} qe_{iq}ife_{iq} \geq 0 \\ (1-q)qe_{iq}ife_{iq} < 0 \end{cases}$, $e_{i,q} = y_i - \mathbf{x}_i'\beta_q$.

The research examines the individual's SWB, which is measured using TOPSIS. The factors that may explain SWB include country, age, education level, religion, how often someone goes to religious services, income, and overall health. The research focuses on the 25th, 50th (median), and 75th percentiles of the SWB scores. In studies using quantile

regression, this division into three parts is standard because it captures the lower, middle, and upper ranges of the conditional distribution. It keeps the distribution balanced and avoids the issue of having too few data points at the extreme ends [51]. Covariates are shown as dummy variables, where 1 means a condition exists. For example, the dummy variable for female gender is 1 for women and 0 for men. We have also checked for multicollinearity among covariates such as age, education, health status and income. We computed the variance inflation factors (VIFs). As all VIF values are below the accepted threshold (maximum VIF = 3.19), multicollinearity is not an issue in our model.

## 5. Results

### 5.1 Subjective well-being

The Fuzzy-Hybrid TOPSIS approach is applied to the ISSP Social Network module (2017) for six countries: Germany, Denmark, Slovenia, Spain, France, Finland, and Hungary. The analysis was conducted at country and regional levels (NUTS2 and NUTS3).

Table 3 shows the results of both PIS and NIS for each indicator included to measure the SWB. The ideal solutions show the maximum and minimum crisp values. It can be seen that the representative groups for PIS and NIS are territories. The group that represents those who have never felt unhappy is the one that lives in the French overseas departments. The same group also represents those who are satisfied with their lives and find it easy to accomplish their goals. Moreover, the German region of Saarland is the representative group of those who do not consider that family members put much pressure on how they live their lives.

On the other hand, the representative groups for the NIS comprise French, Spanish and Finnish territories. In the Yonne region, citizens are particularly unhappy and depressed. French citizens in the Aube region are completely unsatisfied with their lives. The Spanish autonomous city of Ceuta is represented by those who struggle to achieve their goals. Finally, the Finns of the Uusimaa region experience the most family pressure.

Table 4 shows the SWB results obtained through TOPSIS at the country level. The ranking is in descending order. Thus, the results show that countries appear in descending order of SWB, from high to low. The results do not show area homogeneity, as the two northern EU countries have very different SWB patterns: Denmark has the second-highest SWB, while Finland has the second-lowest. Eastern countries also have similar results, with Slovenia ranking third and Hungary ranking last. Spain is in the middle of the ranking, but its SWB is closer to that of the previous country than to that of the next one. Western EU countries also exhibit heterogeneous SWB: France ranks lowest, while German citizens have the highest level of SWB.

Table S1 in S1 File (in the Appendix) shows the results of the SWB at the regional levels of the countries analysed. The results are sorted in descending order to rank the SWB at the regional level as well. Thus, the regions (or provinces in France) in the first positions of the first column on the left of the table are the areas with the highest SWB values. Meanwhile, the territories with the lowest levels of subjective well-being are in the last positions of the last column on the right.

**Table 3. Positive and negative ideal solution.**

| Item | PIS | Group | Country | NIS | Group | Country |
|------|-----|-------|---------|-----|-------|---------|
| C1 | 92.50 | Overseas departments | FR | 45.50 | Yonne | FR |
| C2 | 92.50 | Overseas departments | FR | 50.00 | Aube | FR |
| C4 | 92.50 | Overseas departments | FR | 29.29 | Ceuta | ES |
| C4 | 92.50 | Saarland | DE | 50.00 | Itae-Uusimaa | FI |

Own elaboration; PIS: Positive Ideal Solution; NIS: Negative Ideal Solution. C1: During the past 4 weeks how often have you felt unhappy and depressed? C2: All things considered, how satisfied are you with your life as a whole nowadays? C3: It is easy for me to accomplish my goals. C4: In general, do your family members put pressure on you about the way you live or organise your personal life? FR: France; DE: Germany; ES: Spain; FI: Finland.

**Table 4. Country subjective well-being.**

| Group | C-SWB |
|-------|-------|
| Germany | 0,694 |
| Denmark | 0,643 |
| Slovenia | 0,628 |
| Spain | 0,608 |
| France | 0,524 |
| Finland | 0,523 |
| Hungary | 0,514 |

Own elaboration; C-SWB: Country subjective well-being.

The German regions where citizens are least depressed, most satisfied with their lives and most likely to achieve their goals without experiencing excessive family pressure are the northern regions of Schleswig-Holstein, Bremen and Mecklenburg-Vorpommern, and the southern regions of Saarland and Bavaria. Although Saxony and Hamburg have the lowest SWB values in Germany, they still rank among the highest in the analysed set.

Conversely, some French territories rank last in the regional SWB ranking. Nevertheless, French territories are spread almost evenly across the entire ranking, probably due to the country's significant fragmentation into provinces. In fact, the territory with the highest SWB value is the French overseas regions (DOM). This result partially reflects the outcome of the positive-ideal solution discussed above.

The Slovenian regions with the highest SWB values are Koroska in the north-east and Gorenjska in the north-west. Conversely, the central region of Zasavska and the south-western region of Obalnokraska have the lowest SWB values in Slovenia. All Spanish regions except Ceuta are in the top half of the regional SWB ranking. La Rioja and the Balearic Islands have the highest SWB values.

The northern EU regions of Denmark and Finland show differences in terms of SWB. Danish regions occupy only positions in the first half of the ranking, with the Region of Northern Jutland the best and the Region of Southern Denmark the worst. The Finnish regions are more heterogeneous, with positions ranging from the top 5 (Åland) to the last position (Uusimaa).

Hungary is at the bottom of the SWB ranking at the country level. Despite this, the capital region (Budapest) and the central Transdanubia region show high levels of SWB. Citizens living in the Great Plain and Southern Transdanubia regions are the most depressed, with greater family pressures, difficulty achieving their goals, and a lack of satisfaction with their lives.

Table 5 shows the elasticity results for each region and indicator. Due to the large number of regions analysed, only the total sample, the top five regions and the bottom five regions are shown for the subjective well-being ranking. The results show that SWB is relatively inelastic with respect to all the indicators, particularly in the French overseas regions. In the Finnish regions of Åland and Åland Islands, and in the French regions of Aube, Landes and Indre, the indicator measuring how easily respondents accomplish their goals appears to most influence subjective well-being. Conversely, in the French region of Val-d'Oise, the SWB indicator is most influenced by interviewees' happiness and family pressures. In the German regions of Saarland and Schleswig-Holstein, the SWB indicator is most influenced by happiness and the ease with which goals are achieved. In the Spanish autonomous community of La Rioja, family pressures and depression are the two factors that most affect SWB.

## 5.2 Socio-economic drivers on subjective well-being

It is now interesting to study the relationship between some socio-economic characteristics and their corresponding values of SWB. A quantile regression model was adopted here to analyse the influence of covariates on low SWB values

**Table 5. Elasticities.**

| Group | Country | C1 | C2 | C3 | C4 |
|---|---|---|---|---|---|
| Total | | 0,587 | 0,549 | 0,718 | 0,531 |
| DOM – Overseas departments | FR | 0,004 | 0,004 | 0,005 | 0,004 |
| Saarland | DE | 0,600 | 0,395 | 0,634 | 0,378 |
| La Rioja | ES | 0,748 | 0,512 | 0,216 | 0,972 |
| Schleswig-Holstein | DE | 0,510 | 0,548 | 0,653 | 0,345 |
| Aland | FI | 0,120 | 0,109 | 0,721 | 0,109 |
| Aube | FR | 0,406 | 0,493 | 0,694 | 0,383 |
| Landes | FR | 0,581 | 0,544 | 0,712 | 0,522 |
| Val-d'Oise | FR | 0,729 | 0,581 | 0,552 | 0,847 |
| Indre | FR | 0,581 | 0,544 | 0,712 | 0,522 |
| Itae-Uusimaa | FI | 0,583 | 0,546 | 0,714 | 0,525 |

Own elaboration; C1: During the past 4 weeks how often have you felt unhappy and depressed? C2: All things considered, how satisfied are you with your life as a whole nowadays? C3: It is easy for me to accomplish my goals. C4: In general, do your family members put pressure on you about the way you live or organise your personal life?; DK: Denmark, FR: France; ES: Spain; DE: Germany; FI: Finland; HU: Hungary; SI: Slovenia.

(25th quantile), median levels (median quantile), and high values of subjective well-being (75th quantile) (Table S2 in S1 File). Regressors are dummy variables that indicate whether a condition is present. For example, if the variable of the age 25–35 is equal to 1, then the corresponding individual is between 25 and 35 years old.

A reference category was selected for each covariate in order to compare the values of the coefficients with respect to the category considered. Thus, the reference categories are selected as follows: Spain, under 24 years old; males, upper-level tertiary, Catholics, attending many times to religious events, the highest income level, and excellent health status. Table S2 in S1 File provides the Koenker and Machado [52] pseudo-$R^2$ values for each quantile. The model accounts for between 39% and 48% of the variation in the SWB indicator across the conditional distribution. These values are similar to those typically observed in quantile regression studies using attitude and SWB data. These values demonstrate that the model captures meaningful differences across the lower, middle and upper parts of the SWB distribution.

Age groups exhibit more homogeneous behaviour, with all coefficients confirming that they have higher SWB than those under 24. When analysing the lowest levels of SWB, it is evident that age is a key positive driver of SWB, with older people tending to have higher SWB values. The same results were obtained for the median and the highest SWB values. Gender is also a key driver of SWB: comparing female and male respondents reveals that, for any SWB value, females have lower SWB than males.

The results regarding education are interesting as they depend on both higher levels of education and SWB. Comparing each category with the highest level of education (those with a Master's or PhD) shows that the coefficients are only significant and positive for those with upper secondary or post-secondary education. These results are consistent across all SWB quantiles.

Regarding religion and attendance at religious services, the results depend more on the level of SWB being analysed. Thus, it appears that Protestants and citizens of other faiths have lower SWB than Catholics at higher levels of SWB. Our results conclusively show that attendance at religious services does not affect SWB levels.

The results on individual income are also very informative. It can be seen that there are no significant differences between those who perceive the highest levels of income and those who perceive average-high levels of income. The significant differences are only observed in the four lower-income categories. In all cases and for the extreme levels of SWB (0.25 and 0.75 quantiles), the results show that these income groups have lower SWB than the highest-income group.

Finally, the individual health status results show that, across the remaining health categories and all SWB levels except the median, the coefficients are significant and negative. Thus, it can be concluded that health status is one of the main drivers of SWB. It is interesting to note that the coefficients are lower when the health status is worse (poor, fair, good and very good). Thus, the more precarious the individual's health status, the lower the SWB indicator.

## 6. Discussion

### 6.1 Differences across European regions

This paper aimed to analyse the differences in SWB between EU countries and regions. The Fuzzy-Hybrid TOPSIS methodology was adopted to address the vagueness in the information provided by respondents to the ISSP questionnaire. Therefore, the answers given by the interviewees were transformed into Triangular Fuzzy Numbers, and afterwards, these numbers were defuzzified to obtain crisp values. Thus, the indicator that measures SWB was calculated through the TOPSIS approach.

The positive and negative ideal solutions offer an initial indication of which group has the highest or lowest subjective well-being values. For instance, the French overseas departments scored maximum points in three of the analysed criteria. These regions have tropical climatic and environmental conditions [53]. According to Cuñado & de Gracia [54], therefore, the high level of happiness of citizens living on these islands may depend on the climate.

The TOPSIS indicator measures SWB at an aggregate level across countries. Germany shows the highest levels among the analysed countries. This can be explained by various factors, such as the German government's family-friendly policies [55] or its effective mental healthcare system [56]. Conversely, Finns and Hungarians have the lowest SWB values. In the Finnish context, these results may be associated with how people recognise, evaluate, and communicate the health of the planet. Lyytimäki and Pitkänen [57] link well-being levels to the ecosystem, and the Finnish government runs a persuasive campaign to raise awareness among citizens of the health and well-being benefits of nature in Finland.

Among the 179 regions analysed, the top positions are occupied mainly by the German and Spanish regions. The Saaarland region is the German region with the highest value of regional subjective well-being. The results do not show a clear capital effect across the countries examined. Paris and Helsinki have some of the lowest scores, with Budapest, Madrid and Ljubljana not far behind. Berlin is the only capital that scored highly. While the capital region usually includes surrounding areas, Berlin is a city-state. This difference in area may explain some of the discrepancies, as the average for a metropolitan area can mask variations between the capital and its suburbs [58]. Most capitals perform poorly, which is consistent with previous studies showing that large cities often have lower well-being than smaller towns or rural areas, despite economic advantages [59,60]. High population density, environmental issues, housing costs and weaker social bonds tend to exacerbate this well-being gap. In Paris, social and spatial inequalities exacerbate these issues. Helsinki may experience Nordic urban issues such as seasonal changes that affect mood.

Another significant result to be highlighted is the difference between the island regions of the countries analysed. Corsica and the Canary Islands have lower levels of SWB than the Balearics. The explanation for this can be found in the different levels of tourism across the islands. Following the work of Smith & Diekmann [61], tourism positively increases the factors linked to the levels of life satisfaction.

### 6.2 SWB socio-economic drivers

The results of the quantile regression models confirm that socio-economic characteristics can influence SWB [62,22,15,20]. In line with Steptoe et al. [20], the results demonstrate that age significantly impacts SWB. Older individuals report higher levels of SWB than young people, so the U-shaped [21] is not confirmed here. This can be attributed to the fact that retired individuals are more likely to have achieved their life goals [63], and having more free time enables them to engage in activities that bring them joy [64].

Another interesting result, driven by the estimates from the quantile regression models, is the difference in SWB across levels of education. The study analyses the differences in the SWB between those with a master or PhD and those with a lower level of education. Having a higher educational qualification has a more negative effect on SWB than having a lower one. This result may be a consequence of what scholars call "Post PhD depression" [65,66]. Religion is also considered a key driver in the study of SWB. The results show an evident positive influence of being a Catholic, compared to other religions, on the SWB. According to Cohen [67], spirituality is an excellent predictor of happiness and quality of life, more relevant for Catholics than Jews.

Income is also considered an important predictor of SWB. According to Thomson et al. [68] results, income changes can significantly impact mental health, especially for people in a weak economic state, as they experience the possible economic stress of unemployment on maintaining their quality of life. The effects are more significant when comparing high incomes with those who benefit from welfare or those who lose income. Thus, there is an income effect on SWB only among low-income citizens. As income increases, this does not affect people's well-being [69].

Finally, the relation between health status and subjective well-being has been analysed. The results replicate what Ngamaba et al. [70] stated, as there is a close relationship between SWB and health status. They conclude that improving citizens' health status is a measure governments must take to enhance their SWB.

## 7. Conclusions

The study analyses differences in SWB across EU regions. The ISSP dataset of the Social Network module (2017) was used. The Fuzzy-Hybrid TOPSIS approach was applied to develop an indicator that measured SWB at the aggregate and individual levels. Then, a quantile regression model was implemented to analyse the effect of socio-economic variables on the SWB.

The results showed that Capital regions have lower SWB scores. Furthermore, tourism can be considered a positive proxy for SWB, as the island regions that invest most in tourism have the highest SWB values. However, the SWB is not only influenced by the territory but also by individual socio-economic characteristics. Older age, an intermediate level of education, male gender, higher income, and excellent individual health status were shown to be positive drivers of SWB.

The research is important for understanding SWB at the regional level and the country's overall performance. It provides robust recommendations for addressing ongoing disparities in people's feelings across different regions. Since individual characteristics affect SWB, programmes aimed at specific groups could be effective. Local stakeholders, thus, should implement plans to support young people, individuals on low incomes and those with health issues. These groups often report lower levels of happiness in different regions.

Like any other work, the study has some limitations. It would be interesting to extend the analysis to all the countries of the European Union. Second, only one wave of the ISSP module (2017) is considered. Future research will focus on the dynamic patterns of the SWB indicator over time. And third, the study implements a quantile regression model considering only three quantiles. It would be interesting to fragment the analysis further, identifying which SWB bands are more influenced by the socio-economic variables used in the study.

## Supporting information

**S1 File.** S1 Table. Regional subjective well-being rankings. This table reports the regional subjective well-being (R-SWB) scores and rankings for all NUTS2 and NUTS3 regions across the seven European countries included in the study, calculated using the Fuzzy-Hybrid TOPSIS approach. S2 Table. Quantile regression estimates. This table presents the estimated coefficients, standard errors, and significance levels of the quantile regression models at the 25th, 50th, and 75th percentiles of the subjective well-being distribution.
(DOCX)

## Author contributions

**Conceptualization:** Alessandro Indelicato, Juan Carlos Martín, Vincenzo Marinello.

**Data curation:** Alessandro Indelicato, Juan Carlos Martín.

**Formal analysis:** Alessandro Indelicato, Juan Carlos Martín.

**Funding acquisition:** Alessandro Indelicato, Juan Carlos Martín.

**Investigation:** Alessandro Indelicato, Juan Carlos Martín.

**Methodology:** Alessandro Indelicato, Juan Carlos Martín.

**Project administration:** Alessandro Indelicato, Juan Carlos Martín.

**Resources:** Alessandro Indelicato, Juan Carlos Martín.

**Software:** Alessandro Indelicato, Juan Carlos Martín.

**Supervision:** Alessandro Indelicato, Juan Carlos Martín.

**Validation:** Alessandro Indelicato, Juan Carlos Martín, Vincenzo Marinello.

**Visualization:** Alessandro Indelicato, Juan Carlos Martín.

**Writing – original draft:** Alessandro Indelicato, Juan Carlos Martín.

**Writing – review & editing:** Alessandro Indelicato, Juan Carlos Martín, Vincenzo Marinello.

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
