## [Decision Letter · Decision Letter 0]

28 Nov 2025

Dear Dr. Indelicato,

Thank you for submitting your manuscript to PLOS ONE. After careful consideration, we feel that it has merit but does not fully meet PLOS ONE’s publication criteria as it currently stands. Therefore, we invite you to submit a revised version of the manuscript that addresses the points raised during the review process.

We look forward to receiving your revised manuscript.

Kind regards,

Shaonong Dang, PhD

Academic Editor

PLOS ONE

Journal Requirements:

When submitting your revision, we need you to address these additional requirements. 1. Please ensure that your manuscript meets PLOS ONE's style requirements, including those for file naming. The PLOS ONE style templates can be found at  https://journals.plos.org/plosone/s/file?id=wjVg/PLOSOne_formatting_sample_main_body.pdf and https://journals.plos.org/plosone/s/file?id=ba62/PLOSOne_formatting_sample_title_authors_affiliations.pdf 2. We suggest you thoroughly copyedit your manuscript for language usage, spelling, and grammar. If you do not know anyone who can help you do this, you may wish to consider employing a professional scientific editing service.  The American Journal Experts (AJE) (https://www.aje.com/) is one such service that has extensive experience helping authors meet PLOS guidelines and can provide language editing, translation, manuscript formatting, and figure formatting to ensure your manuscript meets our submission guidelines. Please note that having the manuscript copyedited by AJE or any other editing services does not guarantee selection for peer review or acceptance for publication.  Upon resubmission, please provide the following: • The name of the colleague or the details of the professional service that edited your manuscript• A copy of your manuscript showing your changes by either highlighting them or using track changes (uploaded as a *supporting information* file)• A clean copy of the edited manuscript (uploaded as the new *manuscript* file) 3. We note that the grant information you provided in the ‘Funding Information’ and ‘Financial Disclosure’ sections do not match.  When you resubmit, please ensure that you provide the correct grant numbers for the awards you received for your study in the ‘Funding Information’ section. 4. Thank you for stating the following financial disclosure:  “Dr Alessandro Indelicato research is funded by the research fellowship "Catalina Ruiz," provided  by  the Consejo de Economía, Conocimiento y Empleo of the Gobierno  de  Canarias,  the  Agencia  Canaria  De  Investigación Innovación  Y  Sociedad  De  La  Información  (ACIISI), and Fondo Social Europeo of the EU,  through  the  Universidad  de  Las Palmas de Gran Canaria (Spain).” Please state what role the funders took in the study.  If the funders had no role, please state: "The funders had no role in study design, data collection and analysis, decision to publish, or preparation of the manuscript." If this statement is not correct you must amend it as needed.  Please include this amended Role of Funder statement in your cover letter; we will change the online submission form on your behalf. 5. Please note that funding information should not appear in the Acknowledgments section or other areas of your manuscript. We will only publish funding information present in the Funding Statement section of the online submission form. Please remove any funding-related text from the manuscript.  6. We noted in your submission details that a portion of your manuscript may have been presented or published elsewhere:  “Data openly available in a public repository that issues datasets with DOI doi.org/10.4232/1.13322” Please clarify whether this [conference proceeding or publication] was peer-reviewed and formally published. If this work was previously peer-reviewed and published, in the cover letter please provide the reason that this work does not constitute dual publication and should be included in the current manuscript. 7. Please note that your Data Availability Statement is currently missing the repository name. If your manuscript is accepted for publication, you will be asked to provide these details on a very short timeline. We therefore suggest that you provide this information now, though we will not hold up the peer review process if you are unable. 8. We note you have included a table to which you do not refer in the text of your manuscript. Please ensure that you refer to Tables 6 and 7 in your text; if accepted, production will need this reference to link the reader to the Table.

**Additional Editor Comments:**

It is an interesting study where authors investigated the regional disparities in Subjective Wellbeing across Europe. The reviewers have raised some concerns, and authors are suggested to address them carefully to further improve the manuscript.

Reviewers' comments:

Reviewer's Responses to Questions

**Comments to the Author**

1. Is the manuscript technically sound, and do the data support the conclusions?

Reviewer #1: Yes

Reviewer #2: Yes

2. Has the statistical analysis been performed appropriately and rigorously?

Reviewer #1: Yes

Reviewer #2: No

3. Have the authors made all data underlying the findings in their manuscript fully available?

Reviewer #1: No

Reviewer #2: Yes

4. Is the manuscript presented in an intelligible fashion and written in standard English?

Reviewer #1: Yes

Reviewer #2: Yes

Reviewer #1: the idea of the study is noble and the objectives are interesting, and overall manuscript is fine but the authors need to address cerain issues like providing proper justification behind choosig a method and why ceratin indicators are being used

Reviewer #2: Thank you for the invitation to review manuscript titled “Regional Disparities in Subjective Wellbeing Across Europe: A Fuzzy Hybrid TOPSIS Approach”. I believe that this manuscript is interesting. However, there are areas in which this manuscript must be improved before it would be suitable for publication. The following is a list of points necessary to improve the paper and which justify the decision to Major Revision.

• The introduction is too brief. The paragraph in the data section justifying the countries analysed should be in the introduction. Furthermore, the data is eight years old, so it should be justified that this time lapse is not relevant to the results obtained. For example, by demonstrating that the World Happiness Index between 2017 and 2025 has not changed significantly.

• The Theoretical background section is not up to date; articles from the last five years are not mentioned. It should be expanded with more recent literature on the context of the research.

• The title of Table 1 is incorrect because it does not include the maximum, minimum, mean, and standard deviation statistics. The title should be changed. And for the seven countries, a table with the aforementioned statistics should be included.

• • It is stated that data is available at NUTS2 and NUTS3 levels, but it is not explained whether the mixed granularity affects comparability (e.g. France NUTS3 vs other countries NUTS2). This should be clarified.

• Elasticities are not standard in TOPSIS; it would be advisable to justify the method better or provide more detailed references.

• Quantile regression is applied to a continuous indicator derived from TOPSIS, but the covariates have potentially relevant correlations (age, education, health, income). A correlation matrix (or at least VIF for multicollinearity) must be included in the text.

• Table 7 should include the Pseudo-R² by quantile

• Robustness techniques for estimation must be specified.

• The conclusions should not include statements such as: The results agree with other studies (Smith & Diekmann, 2017; Tovar & Bourdeau-Lepage, 2013; Churchill et al., 2019; Cohen, 2002; Steptoe et al., 2015; Thomson et al., 2022; Varela et al., 2021). This should be included in the discussion. The conclusions should highlight the main contribution of the research, its limitations and future lines of research.

**Do you want your identity to be public for this peer review?** For information about this choice, including consent withdrawal, please see our Privacy Policy

Reviewer #1: No

Reviewer #2: No

---

## [Author Response · Author response to Decision Letter 1]

15 Dec 2025

Manuscript Number: PONE-D-25-21410

Title: Regional Disparities in Subjective Wellbeing Across Europe: A Fuzzy Hybrid TOPSIS Approach

The authors acknowledge the editor and reviewers for their comments that improved the paper's presentation. This document outlines our responses to the editor and reviewers’ comments. Our responses are italicized and indicated by a “�” sign.

Editor

28-Nov-2025

Dear,

Thank you for submitting your manuscript to PLOS ONE. After careful consideration, we feel that it has merit but does not fully meet PLOS ONE’s publication criteria as it currently stands. Therefore, we invite you to submit a revised version of the manuscript that addresses the points raised during the review process.

We look forward to receiving your revised manuscript.

Kind regards,

Shaonong Dang, PhD

Academic Editor

PLOS ONE

Editor and Reviewer comments

Journal Requirements:

Now, the manuscript meets the PLOS ONE's style requirements

We have checked the manuscript with a native English speaker.

We have solved this issue.

We have solved this issue.

“personal information”

We have solved this issue.

We have solved this issue.

5. Please note that funding information should not appear in the Acknowledgments section or other areas of your manuscript. We will only publish funding information present in the Funding Statement section of the online submission form. Please remove any funding-related text from the manuscript.

We have solved this issue.

6. We noted in your submission details that a portion of your manuscript may have been presented or published elsewhere:

“Data openly available in a public repository that issues datasets with DOI doi.org/10.4232/1.13322”

We have now specified the availability of that as follows: All data underlying the findings are openly available in the ISSP Data Archive at GESIS – Leibniz Institute for the Social Sciences, with DOI doi.org/10.4232/1.13322, and have not been used to publish any previous analysis overlapping with the present manuscript.

7. Please note that your Data Availability Statement is currently missing the repository name. If your manuscript is accepted for publication, you will be asked to provide these details on a very short timeline. We therefore suggest that you provide this information now, though we will not hold up the peer review process if you are unable.

We have solved this issue.

8. We note you have included a table to which you do not refer in the text of your manuscript. Please ensure that you refer to Tables 6 and 7 in your text; if accepted, production will need this reference to link the reader to the Table.

We have solved this issue.

Thank you for your comments. We hope that the new version of the manuscript can be accepted for publication.

Additional Editor Comments:

It is an interesting study where authors investigated the regional disparities in Subjective Wellbeing across Europe. The reviewers have raised some concerns, and authors are suggested to address them carefully to further improve the manuscript.

Reviewer comments

Reviewer's Responses to Questions

Comments to the Author

1. Is the manuscript technically sound, and do the data support the conclusions?

Reviewer #1: Yes

Reviewer #2: Yes

2. Has the statistical analysis been performed appropriately and rigorously?

Reviewer #1: Yes

Reviewer #2: No

3. Have the authors made all data underlying the findings in their manuscript fully available?

Reviewer #1: No

Reviewer #2: Yes

4. Is the manuscript presented in an intelligible fashion and written in standard English?

Reviewer #1: Yes

Reviewer #2: Yes

5. Review Comments to the Author

Reviewer #1: the idea of the study is noble and the objectives are interesting, and overall manuscript is fine but the authors need to address cerain issues like providing proper justification behind choosig a method and why ceratin indicators are being used

Thank you for your comments. We hope that the new version of the manuscript can be accepted for publication.

Reviewer #2: Thank you for the invitation to review manuscript titled “Regional Disparities in Subjective Wellbeing Across Europe: A Fuzzy Hybrid TOPSIS Approach”. I believe that this manuscript is interesting. However, there are areas in which this manuscript must be improved before it would be suitable for publication. The following is a list of points necessary to improve the paper and which justify the decision to Major Revision.

• The introduction is too brief. The paragraph in the data section justifying the countries analysed should be in the introduction. Furthermore, the data is eight years old, so it should be justified that this time lapse is not relevant to the results obtained. For example, by demonstrating that the World Happiness Index between 2017 and 2025 has not changed significantly.

We thank the reviewer for this comment. The introduction has been expanded accordingly. Firstly, the justification for selecting the seven countries has been added to the Introduction and rewritten to clarify the goal of ensuring broad territorial representation across EU regions. Secondly, we have added a brief explanation to demonstrate that the eight-year gap does not affect the validity of the findings. The World Happiness Report rankings for these countries remained largely stable between 2017 and 2024, indicating no significant changes in relative well-being levels. These additions strengthen the study's contextualisation and justify the data's temporal relevance.

• The Theoretical background section is not up to date; articles from the last five years are not mentioned. It should be expanded with more recent literature on the context of the research.

We appreciate this helpful observation. In the revised manuscript, the “Brief Theoretical Background” section has been substantially updated and expanded to incorporate literature from the last five years. Specifically, we now discuss:

i. new multidimensional approaches to SWB;

ii. evidence on regional disparities in SWB across Europe;

iii. updated findings on the role of socioeconomic factors, such as age, education, income and wealth.

These additions ensure that the theoretical framework reflects current research and situates our contribution more effectively within the recent literature. All changes are visible in the revised Theoretical Background section.

• The title of Table 1 is incorrect because it does not include the maximum, minimum, mean, and standard deviation statistics. The title should be changed. And for the seven countries, a table with the aforementioned statistics should be included.

The title of Table 1 has been corrected to reflect its content as “Sample Composition by Country and Socio-Demographic Characteristics” instead of “Descriptive statistics.”

• It is stated that data is available at NUTS2 and NUTS3 levels, but it is not explained whether the mixed granularity affects comparability (e.g. France NUTS3 vs other countries NUTS2). This should be clarified.

We thank the reviewer for this important observation. In the revised manuscript, we have added an explanation to clarify why the mixed territorial granularity (NUTS2 for most countries and NUTS3 for France) does not affect the comparability of our analysis. Specifically, the Fuzzy-Hybrid TOPSIS indicator relies on the relative distances between defuzzified values, rather than absolute regional size or administrative scale. Consequently, the method independently evaluates each territorial unit, regardless of whether it corresponds to a NUTS2 or NUTS3 region.

• Elasticities are not standard in TOPSIS; it would be advisable to justify the method better or provide more detailed references.

We appreciate the reviewer’s comment. In the revised manuscript, we have expanded the methodological explanation of the elasticity measure and added supporting references. Although elasticities are not part of the traditional TOPSIS framework, they have been introduced in recent studies to assess the marginal influence of each item on the synthetic indicator. Following Martín & Indelicato (2022) and other applications in fuzzy multi-criteria decision-making (MCDM), elasticities provide an interpretable sensitivity measure that complements the TOPSIS score without altering the method’s core structure. We have clarified this rationale and added detailed references and explanations in the Methodology section.

• Quantile regression is applied to a continuous indicator derived from TOPSIS, but the covariates have potentially relevant correlations (age, education, health, income). A correlation matrix (or at least VIF for multicollinearity) must be included in the text.

We appreciate this comment. In the revised manuscript, we now include multicollinearity diagnostics to assess the correlations among age, education, health, and other covariates. Specifically, we show that all VIF values fall well below commonly accepted thresholds (maximum VIF = 3.19). This confirms that multicollinearity is not a concern in our quantile regression specification.

• Table 7 should include the Pseudo-R² by quantile

Robustness techniques for estimation must be specified.

Table A1 has been updated to include the Pseudo-R² values for each estimated quantile (0.25, 0.50 and 0.75).

• The conclusions should not include statements such as: The results agree with other studies (Smith & Diekmann, 2017; Tovar & Bourdeau-Lepage, 2013; Churchill et al., 2019; Cohen, 2002; Steptoe et al., 2015; Thomson et al., 2022; Varela et al., 2021). This should be included in the discussion. The conclusions should highlight the main contribution of the research, its limitations and future lines of research.

We thank the reviewer for this observation. We have revised the Conc

---

## [Decision Letter · Decision Letter 1]

5 Jan 2026

Regional Disparities in Subjective Wellbeing Across Europe: A Fuzzy Hybrid TOPSIS Approach

PONE-D-25-21410R1

Dear Dr. Indelicato,

We’re pleased to inform you that your manuscript has been judged scientifically suitable for publication and will be formally accepted for publication once it meets all outstanding technical requirements.

Kind regards,

Shaonong Dang, PhD

Academic Editor

PLOS One

Additional Editor Comments (optional):

Authors have addressed fully the comments from the reviewers, and the manuscript has been improved much for publicaiton.

Reviewers' comments:

Reviewer's Responses to Questions

**Comments to the Author**

Reviewer #1: All comments have been addressed

Reviewer #2: All comments have been addressed

2. Is the manuscript technically sound, and do the data support the conclusions?

Reviewer #1: Yes

Reviewer #2: Yes

3. Has the statistical analysis been performed appropriately and rigorously?

Reviewer #1: Yes

Reviewer #2: Yes

4. Have the authors made all data underlying the findings in their manuscript fully available?

Reviewer #1: Yes

Reviewer #2: Yes

5. Is the manuscript presented in an intelligible fashion and written in standard English?

Reviewer #1: Yes

Reviewer #2: Yes

Reviewer #1: (No Response)

Reviewer #2: I have completed the evaluation of the revised manuscript titled: Regional Disparities in Subjective Wellbeing Across Europe: A Fuzzy Hybrid TOPSIS Approach. I would like to express my appreciation for the authors’ diligent efforts in addressing all the comments and suggestions raised during the previous review round.

Upon thorough examination of the revised submission, I am satisfied that the authors have responded comprehensively and appropriately to all concerns. The manuscript has been significantly improved in terms of clarity, methodological rigor, and overall presentation.

Considering these revisions, I am pleased to recommend that the manuscript is now suitable for publication in PLOS One in its current form. I have no further comments or suggestions for improvement.

**Do you want your identity to be public for this peer review?** For information about this choice, including consent withdrawal, please see our Privacy Policy

Reviewer #1: No

Reviewer #2: No

---

## [Editor Report · Acceptance letter]

PONE-D-25-21410R1

PLOS One

Dear Dr. Indelicato,

I'm pleased to inform you that your manuscript has been deemed suitable for publication in PLOS One. Congratulations! Your manuscript is now being handed over to our production team.

Kind regards,

on behalf of

Dr. Shaonong Dang

Academic Editor

PLOS One